# Antioxidant and Anticancer Potential of the New Cu(II) Complexes Bearing Imine-Phenolate Ligands with Pendant Amine N-Donor Groups

**DOI:** 10.3390/pharmaceutics15020376

**Published:** 2023-01-22

**Authors:** Adriana Castro Pinheiro, Ianka Jacondino Nunes, Wesley Vieira Ferreira, Paula Pellenz Tomasini, Cristiano Trindade, Carolina Cristóvão Martins, Ethel Antunes Wilhelm, Robson da Silva Oliboni, Paulo Augusto Netz, Rafael Stieler, Osvaldo de Lazaro Casagrande, Jenifer Saffi

**Affiliations:** 1Laboratory of Genetic Toxicology, Department of Basic Health Sciences, Federal University of Health Sciences of Porto Alegre (UFCSPA), Porto Alegre 90050-170, RS, Brazil; 2Group of Catalysis of Theoretical Studies, Center of Chemical, Pharmaceutical and Food Science Center, Federal University of Pelotas (UFPel), Pelotas 96160-000, RS, Brazil; 3Centro de Investigaciones en Ciencias de la Vida, Universidad Simón Bolívar, Barranquilla 080002, Colombia; 4Laboratory in Biochemical Pharmacology, Center of Chemical, Pharmaceutical and Food Sciences, Federal University of Pelotas (UFPel), Pelotas 96160-000, RS, Brazil; 5Grupo de Química Teórica, Instituto de Química, Universidade Federal do Rio Grande do Sul (UFRGS), Porto Alegre 91501-970, RS, Brazil; 6Laboratory of Molecular Catalysis, Instituto de Química, Universidade Federal do Rio Grande do Sul (UFRGS), Porto Alegre 91501-970, RS, Brazil

**Keywords:** cancer, copper complexes, Schiff base, antioxidant activities, cytotoxicity, DNA damage, DNA-targeting agent

## Abstract

Cu(II) complexes bearing NNO-donor Schiff base ligands (**2a**, **b**) have been synthesized and characterized. The single crystal X-ray analysis of the **2a** complex revealed that a mononuclear and a dinuclear complex co-crystallize in the solid state. The electronic structures of the complexes are optimized by Density Functional Theory (DFT) calculations. The monomeric nature of **2a** and **2b** species is maintained in solution. Antioxidant activities of the ligands (**1a**, **b**) and Cu(II) complexes (**2a**, **b**) were determined by in vitro assays such as 1,1-diphenyl-2-picrylhydrazyl free radicals (DPPH^.^) and 2,2′-azino-bis (3-ethylbenzothiazoline-6-sulfonic acid) radicals (ABTS^+^). Our results demonstrated that **2a** showed better antioxidant activity. MTT assays were performed to assess the toxicity of ligands and Cu(II) complexes in V79 cells. The antiproliferative activity of compounds was tested against two human tumor cell lines: MCF-7 (breast adenocarcinoma) and SW620 (colorectal carcinoma) and on MRC-5 (normal lung fibroblast). All compounds showed high cytotoxicity in the all-cell lines but showed no selectivity for tumor cell lines. Antiproliferative activity by clonogenic assay **2b** showed a more significant inhibitory effect on the MCF-7 cell lines than on MRC-5. DNA damage for the **2b** compound at 10 µM concentration was about three times higher in MCF-7 cells than in MRC-5 cells.

## 1. Introduction

Metallodrugs are compounds that have promising prospects for meeting biomedical challenges. These pharmaceuticals have been applied successfully in antineoplastic chemotherapy for different cancers [1,2,3,4,5,6,7,8]. Cisplatin, carboplatin, and oxaliplatin are coordination compounds approved worldwide for clinical use in cancer treatment [9,10,11,12,13]. The cytotoxic activity of these metal complexes is attributed to interactions with cellular targets, such as binding to genomic DNA sequences and interaction with mitochondrial DNA and proteins [10,14]. However, these anticancer inorganics compounds have associated adverse effects that negatively impact the quality of life and inherent or acquired resistance [15,16]. Such limitations drive research in identifying novel anticancer metal complexes and therapeutic strategies to overcome the problem of chemoresistance and toxicity of the compounds currently in clinical use.

Copper is among the most examined metal ions due to its significant role in cancer [17,18]. Abnormal Cu levels in cancer cells have become a new target for cancer treatment [19,20,21]. Cu is involved in cell proliferation, angiogenesis, and metastasis, which are essential processes of cancer progression [22]. Typical Cu-induced tumor cell death mechanisms include oxidative stress [23], proteasome [23,24] and DNA Topoisomerase inhibition [25,26,27], antiangiogenesis [23], and cuproptosis [28,29]. Therefore, cancer cells may represent a selective target for copper-based agents. In this way, the use of Cu chelators reduces the Cu concentration in tumor cells [30], and ionophores increase intracellular Cu levels [31]. The mechanism of action of chelators and ionophores is based on the formation of metal complexes. Copper complexes exhibit a differential response to tumor cells as compared with normal cells [32]. Furthermore, it is thought to be less toxic than nonessential metals such as platinum and is considered a promising alternative to platinum compounds [19]. The properties of copper complexes are determined mainly by the nature of the ligands present in the complexes. The ligands can modulate essential aspects of the metal complexes’ pharmacological activity, changing parameters such as lipophilicity, solubility, stability, and metabolism in biological systems [33]. Thus, synergistic effects between the metal and the ligand enhance efficiency and reduce toxic side effects, providing better therapeutic effects.

Schiff bases containing azomethine group (–CH=N–) are scaffolds of high interest in medicinal chemistry due to their interesting pharmacological properties, thermal stability, high synthesis, flexibility, and chelating ability with metal ions [34,35,36,37,38]. Cu(II) complexes supporting Schiff bases ligands have shown antiproliferative and antitumor activity [24,38,39,40], anti-inflammatory [41], antioxidant [42], and antibacterial antimicrobial activity [43,44].

Cells in tumors present higher levels of ROS and are particularly sensitive to oxidative stress. This way, antioxidants are being explored to find more effective cancer therapies. Antioxidants can minimize the burden of free reactive radicals in cells caused by chemotherapeutic agents [45]. So, it can decrease the duration of chemotherapy regimens and reduce the side effects caused by the treatment.

Based on these facts, we are interested in determining the impact of amine moiety changes (both electronic and rigid vs. flexible) on the biological activity of the two copper complexes containing Schiff base chromophore and phenoxide fragment at the second and third coordination positions, respectively.

In particular, quinoline is a heterocyclic organic compound with the chemical formula C_9_H_7_N. This compound shows weak tertiary base properties because the nitrogen atom in the structure pulls electrons by resonance [46,47]. Anticancer drugs used clinically, such as Camptothecin, Topotecan, Irinotecan, etc., highlight the importance of the quinoline framework in anticancer drug development [41]. These anticancer-agent quinoline derivatives have been described with diverse mechanisms of action such as topoisomerase I and II inhibitors, iron chelators, carbonic anhydrase inhibitors, DNA intercalating agents, telomerase inhibitors, G-quadruplex-DNA binding affinity, angiogenesis inhibitors, etc. [48].

Thus, we have synthesized and characterized two Cu(II) complexes chelated by N,N,O-tridentate Schiff base ligands. Following this research line, biological studies of these complexes were carried out in vitro for antioxidant activity, cytotoxicity, antiproliferative, and genotoxicity studies. The observed biological activity in human tumor cell lines has been correlated with the differences in the resulting chemical structure to obtain some structure-activity relationship information.

## 2. Materials and Methods

### 2.1. Chemicals and Physical Measurements

All reagents and solvents were obtained from commercial suppliers (Aldrich, Merck, and Flucka), used as received, and of the highest purity. Infrared spectra were performed on an FT-IR Bruker Alpha Spectrometer operating in the ATR mode. NMR spectra of the ligands (**1a**, **b**) were recorded on Varian MR 400 MHz spectrometer operating at 25 °C. Elemental analyses were performed by the Analytical Central Service of the Chemistry Institute—UFRGS (Porto Alegre, Brazil) and are the average of two independent determinations. High-resolution mass spectra (HRMS) of Cu(II) complexes (**2a**, **b**) were obtained by electrospray ionization (ESI) in a positive mode in CH_3_OH solutions using a Micromass Waters^®^ Q-Tof spectrometer. The electronic absorption spectra of the Cu(II) complexes were recorded using Cary 60 Spectrophotometer from 200 to 1200 nm. Cyclic voltammograms (CV) were measured using a Potentiostat/Galvanostat AUTO LAB PGSTAT 30/FRA 2 at room temperature. These CV experiments were carried out by employing a standard three-component system: a platinum planar electrode as the working electrode, a platinum wire auxiliary electrode, and a saturated calomel electrode (SCE) as the reference electrode. All scans were recorded at 100 mVs^−1^ scan rate in DMF containing 1 × 10^−2^ M complexes and 0.1 M LiClO_4_ as a supporting electrolyte. The ferrocene/ferrocenium couple was used as a standard. All potentials are referenced to the SCE reference electrode.

### 2.2. Synthesis

#### 2.2.1. Synthesis of the Ligands 

[6-(Ph(NH)-C_2_H_4_-(N=CH)-2,4-di-*tert*-butyl-(OHC_6_H_2_)] (**1a**), and [6-(C_9_H_6_N-8-(N=CH))-2,4-di-*tert*-butyl-(OHC_6_H_2_)] (**1b**) ligands were prepared according to the procedure described in [49,50].

#### 2.2.2. Synthesis of Copper Complexes

Synthesis of copper complex **2a***:* For the synthesis of **2a**, we used a methanolic solution (10 mL) of **1a** (0.155 g, 0.44 mmol) at 40 °C, which was added dropwise to a solution of CuCl_2_·2H_2_O (0.075 g, 0.44 mmol) in methanol (5 mL). The reaction mixture was left stirring for 3 h at room temperature. After completion of the reaction, the resulting solution was concentrated (ca. 2 mL) to give **2a** as dark green crystals, after a few days. A single crystal proved suitable for X-ray diffraction studies. Yield: 82%. Anal. Calcd. for C_23_H_32_CuN_2_O (**2a**): C 61.32; H 6.94; N 6.22. Found: C 60.91; H 6.75; N 6.31. ESI-HRMS (MeOH, *m/z*) = 414.1711 [M-Cl]^+^ (calcd. for C_23_H_31_CuN_2_O: 414.1732). UV-Vis (CH_2_Cl_2_, nm): 229, 266, 290, 348, 647. UV-Vis (DMSO, nm): 279, 312, 382, 675.

Synthesis of copper complex ***2b***: This complex was prepared as described above for **2a**, starting from **1b** (0.200 g, 0.55 mmol) and CuCl_2_·2H_2_O (0.094 g, 0.55 mmol) to give **2b** as a brown solid. Yield: 87%. Anal. Calcd. for C_24_H_27_ClCuN_2_O (**2a**): C 62.87; H 5.94; N 6.11. Found: C 61.38; H 6.23; N 5.95. ESI-HRMS (MeOH, *m/z*) = 422.1416 [M-Cl]^+^ (calcd. for C_24_H_27_CuN_2_O: 422.1419). UV-Vis (CH_2_Cl_2_, nm): 237, 288, 348, 483. UV-Vis (DMSO, nm): 260, 346, 472.

### 2.3. Single Crystal X-ray Diffraction

A Bruker D8 Venture Photon 100 dual source diffractometer was used to collect X-ray data for structural analysis. Data were collected using Mo-Kα radiation and a combination of ϕ and ω scans was carried out to obtain at least one unique data set. The crystal structures were solved using direct methods with SHELXS [51]. The final structures were refined using SHELXL [52], where the remaining atoms were located by difference Fourier synthesis. Anisotropic displacement parameters were applied to all non-hydrogen atoms followed by full-matrix least-squares refinement based on F2. All hydrogen atoms were placed in ideal positions and refined as riding atoms with relative isotropic displacement parameters. Drawings were made using the program Diamond (version 4.6.0) [53].

### 2.4. Theoretical Methods

Geometry optimizations of the free ligands **1a**, **b** and its Cu(II) complexes **2a**, **b** were performed using the ORCA program package [54]. Density Functional Theory (DFT) calculations were performed using the B3LYP hybrid functional [55,56] together with the Ahlrichs-type basis set TZVPP for the metal and all its coordinated atoms and SVP for other atoms were used [57,58,59], combined with the def2/J auxiliary basis [60]. The resolution of identity approximation [61] and Grimme’s dispersion correction [62,63] were used throughout all calculations. X-ray diffraction coordinates were used to calculate the geometry optimization of the **2a**.

### 2.5. Molecular Docking

Docking analysis was performed with AutoDock Vina [64] using B-DNA (PDB: 1BNA [65]) as a target. The crystal structure of the receptors was retrieved from the Protein Data Bank and prepared with Autodock tools and UCS Chimera [66]. Nucleotide-ligand interactions were calculated with the Protein–Ligand Interaction Profiler (PLIP) [67] and the Discovery Studio 2021 software [68].

### 2.6. Studies on the Stability of the Complexes in Solution

The behavior/stability in the solution of complexes **2a**, **b** were studied for 24 h in DMSO and for 72 h in H_2_O by UV-Vis spectroscopy. The stock solutions of the test compounds were prepared in DMSO and then diluted to a final concentration of 1 × 10^−7^ M with H_2_O (10% DMSO). The absorbance spectra of samples were recorded in the range of 250–1000 nm.

### 2.7. Biological Activities Studies

#### 2.7.1. Determination of Free Radical Scavenging Capacity

The free radical scavenging ability of the compounds (**1a**, **b** and **2a**, **b**) was tested by 1,1-diphenyl-2-picrylhydrazyl (DPPH) and 2,2′-azinobis-(3-ethylbenzothiazoline)-6-sulfonic acid (ABTS) radical scavenging assay [69,70] with some modifications. Both colorimetric assays are being used to estimate the radical scavenging capacity of organic and inorganic compounds [71,72]. When DPPH reacts with an antioxidant compound, its free radical property is lost, and its color changes from violet to yellow. In the presence of an antioxidant, the ABTS radical, which has a strong absorption band at 734 nm, will be scavenged, and the reaction mixture will turn from blue to green. The results are expressed as a percentage of inhibition of DPPH radical or ABTS^+^ radical when compared to samples from the control group (DMSO) without the compounds/standard (0% of radical inhibition). The experiment was repeated three times at each concentration.

DPPH assay: To determine the 1,1-diphenylpicrylhydrazyl (DPPH) radical scavenging activity, different concentrations of compounds (1, 5, 10, 50, 100, and 200 μM) were mixed with a methanolic solution of 0.1 mM DPPH radical for 30 min at 30 °C in the dark. The absorbance of the mixture was measured spectrophotometrically at 517 nm. The coloration change of the DPPH ethanolic solution from deep purple to yellow indicates the presence of antioxidants and then the free radical scavenging effect of these compounds. The ascorbic acid at 1, 10, 100, and 200 µM was used as a positive control to determine the maximal decrease in the DPPH absorbance. The percentage of DPPH radical scavenging activity was calculated by the following equation:% DPPH radical scavenging activity = {(A_0_ − A_1_)/A_0_} × 100
where A_0_ is the absorbance of the control, and A_1_ is the absorbance of the compounds/standard. 

ABTS assay: The ligands (**1a**, **b**) and Cu(II) complexes (**2a**, **b**) were assayed at 1, 5, 10, 50, 100, and 200 µM in the ABTS assay. The radical cation ABTS was previously generated with sodium persulfate in the presence of potassium phosphate buffer. Briefly, the ABTS solution diluted in a phosphate buffer saline pH 7.4 was mixed with an aliquot of the compound for 30 min at room temperature in the dark. The ascorbic acid at 1, 10, 100, and 200 µM was used as a positive control to determine the maximal decrease in the ABTS^+^ absorbance. The color reaction was measured at 730 nm and the results were expressed as a percentage (%) of the control. The ABTS radical scavenging capacity was calculated by the following equation: % ABTS^+^ radical scavenging activity = {(A_0_ − A_1_)/A_0_} × 100
where A_0_ indicates the absorbance of the blank and A_1_ indicates the absorbance in the presence of the test sample.

#### 2.7.2. Cell Culture and Treatments

Chinese hamster lung fibroblast (V79) cells and human cell lines [MCF-7 (breast adenocarcinoma), SW620 (colorectal adenocarcinoma), and MRC-5 (normal lung fibroblast)] were obtained from the Rio de Janeiro Cell Bank (Rio de Janeiro, RJ, Brazil). The cell lines were grown in DMEM supplemented with 10% heat-inactivated FBS or RPMI-1640 supplemented with 20% FBS, 0.2 mg/mL L-glutamine, 100 IU/mL penicillin, and 100 µg/mL streptomycin. Cells were kept in tissue culture flasks at 37 °C in a humidified atmosphere containing 5% CO_2_ and were harvested by treatment with 0.15% trypsin −0.08% EDTA in PBS. 

#### 2.7.3. Cell Viability Assessment by MTT Assay

The cytotoxic potential of all compounds was measured by the methylthiazol tetrazolium (MTT) assay in Chinese hamster lung fibroblasts and human cell lines. Cell lines were cultured in a 96-well plate (1 × 10^4^ cells/well) in complete media. V79 cells were treated for 24 h with all compounds at concentrations of 1, 10, 50 and100 µM. For human cells, concentrations of 1, 10, 20, 30, 40, 50, 75, 85, and 100 µM of the ligands (**1a**, **b**) and Cu(II) complex **2a** for 24 h and 72 h were applied. Treatment with complex **2b** evaluated concentrations of 1, 2, 4, 10, 20, and 30 μM for 24 h and 1, 2, 3, 4, 5, 7.5, 8.5, and 10 μM for 72 h in all human cell lines. All compounds were dissolved in DMSO and added to the medium to reach the desired concentrations (0.1% of DMSO). The activity of this solvent alone was measured by a blank sample solution containing the same amount of DMSO. After the exposure period, the culture media were replaced with the addition of 100 µL serum-free media containing the 3-(4,5-dimethylthiazol-2-yl)-2,5-diphenyltetrazolium bromide) (MTT) dye (1 mg/mL), and incubated for 3 h at 37 °C After incubation, the supernatant was removed, the residual purple formazan product was solubilized in 150 µL DMSO, and its absorbance was measured at 570 nm. The assay was performed in triplicate. Cell viability (%) and IC_50_ (drug concentration required to inhibit the cell growth by 50% after 24 h or 72 h of incubation) were obtained through GraphPad Prism 5 (GraphPad Software, Inc., San Diego, CA, USA). 

#### 2.7.4. Clonogenic Assay

MCF-7, MRC-5, and SW620 human cell lines were seeded in 6-well plates at a density of 300 cells per well containing 1 mL of medium. After 24 h of incubation, cells were treated with the ligands **1a** and **1b** (1, 10, and 20 µM) and Cu(II) complexes **2a** (1, 10, and 20 µM) and **2b** (1 and 4 μM) for 72 h. The cells were cultured in a complete medium without compounds **2a** and **2b** (control) for another 8 days for the MRC-5 strain, 10 days for SW620, and 14 days for MCF-7. The cell culture medium was changed to a new medium every 2 days. Cells were washed twice with PBS and fixed with methanol for 30 min, followed by staining with 0.5% crystal violet. The assay was performed in triplicate. Colonies were manually counted, and colony survival was presented as a percentage relative to the respective negative control.

#### 2.7.5. Alkaline Comet Assay

For genotoxicity assays, 1.0 × 10^4^ cells were seeded on twenty-four-well plates and incubated for 24 h. Cells were incubated with various concentrations of **1a**, **b** and **2a**, **b** for 3 h in an FBS-free medium. Each dose was evaluated by three independent experiments. The alkaline comet assay was performed as described by Singh et al. [73]. Following treatment, cells were trypsinized and resuspended in a complete medium. Slides were prepared with 20 μL of cellular suspension and were dissolved in 0.75% low melting-point agarose. Slides were immersed in cold lysis buffer (2.5 M NaCl, 100 mM EDTA, 10 mM Tris, 1% Triton X-100, 10% DMSO, pH 10) for at least 1 h. Slides were placed in an electrophoresis chamber with cold alkaline buffer (300 mM NaOH and 1 mM de EDTA, pH > 13) to unwind the DNA. Electrophoresis was performed at 0.7 V/cm, 300 mA for 30 min at 4 °C. Slides were then neutralized with Tris–HCl buffer (pH 7.5) for 15 min and stored at room temperature. Slides were stained with silver staining protocol. The gels were then left to dry overnight at room temperature before being analyzed. From two replicate slides, 100 cells were selected and analyzed for DNA migration. Migration of DNA fragments was determined according to comet class as described by Azqueta [74]: class 0, intact nuclei, without tail; class 1, nuclei with tail less than the diameter of the nucleus; class 2, tail size varying between one and two times the diameter of the nucleus; class 3, tail size varying between two and three times the diameter of the nucleus; and class 4, tail size more than three times the diameter of the nucleus but with the head and tail of the comet still distinguishable. The DI ranges from 0 (100 completely undamaged cells × 0) to 400 (100 cells with maximum damage × 4). 

### 2.8. Statistical Analysis 

Statistical analyses were performed using GraphPad Prism 5 (GraphPad Software, Inc.). For comparisons, ANOVA followed by Tukey’s post-test or ANOVA followed by Dunnett’s multiple comparison test or ANOVA and Bonferroni post-test to compare replicate means of each tumor cell line with MRC-5 cell line were used. A *p*-value < 0.05 was considered statistically significant. 

## 3. Results and Discussion

### 3.1. Chemical Section

#### 3.1.1. Synthesis and Characterization of the Compounds

The tridentate iminophenolate ligands (**1a**, **b**) with pendant N-donor groups were synthesized by Schiff base condensations between the corresponding primary amine and 2,4-di-*tert*-butylsalicylaldehyde in refluxing methanol and their identities were confirmed by NMR spectroscopy (Appendix A) [49,50]. The equimolar reaction of **1a** and **1b** with CuCl_2_·2H_2_O in methanol for 3 h at room temperature yielded two new Cu(II) complexes as dark green (**2a**) and brown (**2b**) solids in very good yields (Figure 1). These complexes are readily soluble in common organic solvents such as tetrahydrofuran, toluene, and dichloromethane at room temperature; however, the solubility of the **2b** in these solvents is very limited. Particularly, better solubility of **2b** can be reached using coordinating solvents such as dimethyl sulfoxide (DMSO). Both **2a** and **2b** are slightly soluble in an aqueous solution.

Complexes **2a** and **2b** were characterized in the solid state and solution by elemental analysis, UV-Vis spectroscopy, and high-resolution mass spectrometry (HRMS). The elemental analysis results agree with those calculated values, confirming the purity of these compounds. Unfortunately, **2a** and **2b** showed paramagnetic nature, and informative NMR data cannot be obtained.

The ESI-HRMS results indicated that the monomeric nature of **2a** and **2b** species is maintained in solution with the formation of [M–Cl]^+^ ions for **2a** (*m/z* = 414.1711), and [M–Cl]^+^ ions for **2b** (*m/z* = 422.1416) (Appendix A). UV–visible spectra of the tridentate Schiff base ligands and copper complexes were recorded in dichloromethane at room temperature revealed absorption bands for intraligand transition in the range of 234–289 nm which may be associated with π-π* transitions and the low-energy absorptions in the region of ~330–350 nm most likely to be associated with n-π* transitions (see the Appendix A) [75]. In addition, the UV spectra of the **2a** and **2b** showed one absorption peaks around ∼416–480 nm, which can be assigned to the metal-to-ligand charge transfer (MLCT), and one absorption band in the region of 647–672 nm associated with d–d transitions.

#### 3.1.2. Description of the Crystal Structure of Complex **2a**

The crystal structure of the **2a** complex was established by X-ray diffraction studies of a single crystal. The solid-state structure of **2a** is shown in Figure 1, and the main crystallographic data and structure refinement parameters are reported in Appendix A. Bond distances and angles are summarized in Table 1. The X-ray diffraction analysis reveals that a mononuclear (**2a**) and a dinuclear (**2a’**) complexes of **2a** co-crystallize in the solid state. In solution, there are only mononuclear species, according to the ESI-MS studies (see Section 3.1.1). The asymmetric unit consists of a discrete mononuclear complex (Cu1) and one-half of a centrosymmetric dinuclear complex (Cu2). In both cases, the copper centers are chelated by one K^3^-O,N,N-phenoxy-imino-amine ligand, with the O-phenoxy *trans* to the N-amine. In the mononuclear specie, the copper center is tetracoordinated with a distorted quadratic geometry. On the other hand, in the Cu_2_Cl_2_ dinuclear specie, the copper centers are pentacoordinated with distorted square pyramidal geometry.

In the dinuclear complex, the N-amine, N-imino, O-phenoxy, and one Cl-bridge atom occupy the basal positions of the square pyramidal geometry, while one Cl-bridge atom bears the apical position. The Cu–Cl bond distances are 2.2497(5) Å and 2.8452(6) Å, and the Cu–Cl–Cu bond angle is 89.481(17)°. The Cu–Cl bond of 2.8452(6) Å was longer than the sum of the covalent radii (2.31 Å), indicating the existence of weak interactions between the chloride and the copper center [76].

The Cu⋯Cu distance is 3.6111(4) Å. Otherwise, the mononuclear complex has only a short intermolecular contact through hydrogen bond (see Appendix A), with Cu⋯Cu distance of 5.3547(4) Å. Similar findings, i.e., co-crystallization of mononuclear and dinuclear species were already reported for copper(II)-{N-(2-hydroxybenzyl)-N-(2-pyridylmethyl)[(3-chloro)(2-hydroxy)]propylamine} complexes [77].

#### 3.1.3. DFT Calculations

Considering the X-ray diffraction analysis results that showed the presence of mono- and dinuclear species in the solid state, we decided to carry out DFT calculations to obtain further insights about the minimum energy structures of **2a**, **2a’,** and **2b**. All optimized structures for the ground state obtained at the B3LYP/def2-TZVPP/def2-SVP level of theory in the gas phase are shown in Figure 2, and selected geometrical parameters are summarized in Appendix A.

The bond distances in the coordination sphere of the metal center are in agreement with the related copper structures reported in the literature [78,79]. The optimized structures of **2a** and **2b** exhibited a distorted quadratic structure, with Cu–Cl bond distances within 2.219 and 2.235 Å, Cu–N_imine_ 1.937 and 2.000 Å, Cu–O_phenoxy_ 1.954 and 1.923 Å, and Cu–N with a distance of 2.109 and 1.962 Å. The dinuclear complex (**2a’**) has a slightly distorted square pyramid geometry with chloride bridge ligands [80]. The Cu–Cl interaction distance is 2.9 Å, in good agreement with the experimental value of 2.8 Å. This value indicates that the interaction type is predominantly electrostatic [76]. The Cu–Cu distance is 3.417 Å, which is in agreement with the experimental value of 3.611 Å, which is smaller in comparison with reported values for related compounds [76].

DFT calculations showed that the energy difference (Δ*E*) between the dinuclear (**2a’**) and mononuclear (**2a**) complexes is only −7.42 kcal·mol^−1^ [**2a’**: *E* = −3990259.34 kcal·mol^−1^; **2a**: *E* = −3990251.92 kcal·mol^−1^]. Thus, the distance between the copper atom and the bridging Cl ligand obtained experimentally (2.8 Å) and by DFT calculations (2.9 Å) strongly suggest a weak electrostatic interaction between two monomeric structures that affords a more stable dimeric species.

#### 3.1.4. Frontier Molecular Orbital Analysis

Computational methods are valuable for determining the molecular structure, stability, and reactivity of the compounds. B3LYP/def2-TZVPP and def2-SVP levels of theory were used to calculate the energy of the frontier molecular orbitals, the highest occupied molecular orbital (HOMO), and the lowest unoccupied molecular orbital (LUMO) as well as the energy gap between the HOMO–LUMO orbitals (ΔE_HOMO-LUMO_). The HOMO (E_HOMO_) and LUMO (E_LUMO_) energies for the ligands and complexes provide information about energy distribution and energetic behavior. The negative magnitude of E_HOMO_ and E_LUMO_ establishes the stability of compounds [81].

The energy of the HOMO and LUMO orbitals obtained values from DFT calculations for **1a**, **b** and **2a**, **b** are given in Table 2. The energy gap (ΔE_HOMO-LUMO_) of these frontier orbitals is shown in Figure 3. For the HOMO and LUMO orbitals, the positive and negative phases are represented in red and blue, respectively.

E_HOMO_ value is related to the electron-donating ability of the molecule, and E_LUMO_ indicates the electron-accepting ability of the molecule. The higher value of E_HOMO_ implies an increasing ability of the molecule to electron-donating to an acceptor molecule. On the other hand, the higher value of E_LUMO_ determines more electron-accepting ability. Thus, the HOMO orbital acts as an electron density donor while LUMO as an acceptor, and the energy gap, E_HOMO-LUMO_, characterizes the capacity of this intramolecular charge transfer, which is favored when the gap is lower. However, greater stability of the molecule and thus less reactivity in reactions has been related to high energy gap values [81].

In our study, the HOMO and LUMO of the ligands are located in different parts of these molecules. The HOMO orbital in the ligand **1a** is localized on the amine moiety and NH group. The LUMO of **1a** is delocalized on phenyl moiety, O-donor atom, imine group, and slightly on the NH group. On the other hand, in **1b** the charge density on HOMO and LUMO may be delocalized on the whole ligand. In this case, the HOMO orbital with little delocalization on amine and *tert*-butyl group, and LUMO localized on phenoxy moiety. The ligand **1a** shows the highest E_HOMO_ and E_LUMO_ value (as can be seen in Table 2) and consequently higher ΔE_HOMO-LUMO_ (3.977 eV and 3.758 eV to **1a** and **1b**, respectively). Then, from Figure 3 and Table 2, we notice that ligand **1b** is the most stable and the least reactive.

Complexes **2a** and **2b** follow a similar tendency to their respective free ligands **1a** and **1b**. However, the complexes have lower E_HOMO_, E_LUMO_, and ΔE_HOMO-LUMO_ values about the ligands. ΔE_HOMO-LUMO_ to **2a** and **2b** is similar (ΔE_HOMO-LUMO_ = 2.798 eV to **2a** and 2.711 eV to **2b**). E_HOMO_ and E_LUMO_ energy of **2a** is higher than **2b**. Therefore, according to the data in Table 2, **2b** can be less stable and more reactive. The HOMO in **2a** is localized on the Cu(II) center, imine and NH- group and LUMO on phenyl moiety of the phenoxy group, O-donor atom, and imine group. The homo and lumo of **2b** delocalized in the entire molecule but without contribution from the *tert*-butyl group atoms.

#### 3.1.5. Electronic Properties

The chemical properties, such as ionization potential (IP), electron affinity (EA), electronegativity (χ), chemical potential (μ), chemical hardness (η), softness (ς), and index of electrophilicity (ϖ) were also evaluated by the DFT method. Table 3 provides the values obtained from these descriptors for **1a**, **b** and **2a**, **b**. These electronic descriptors were calculated to explain the reactivity and the biological activities of the synthesized compounds.

Complex **2a** has the lowest ionization energy (IP = 5.392 eV). The values of the hardness of ligands **1a** and **1b** are higher than complexes **2a** and **2b**, while the softness values are lower. The negative chemical potential is indicative of the stability of their compounds. The global electrophilicity index (ω) assesses the lowering of energy due to maximal electron flow between donor and acceptor and has been used to correlate toxicological behavior [82]. The values of ω show that complexes are more electrophilic than ligands. **2b** with ω = 6.264 eV is more eletrophilic than **2a**, ω = 5.698 eV. The higher values of electronegativity and electrophilicity of the **2b** suggest that this compound interacts more efficiently in the biological environment [83].

#### 3.1.6. Studies on the Stability of the Complexes in Solution

The stability of pharmaceutical products is an important test of their approbation. Then, electronic absorption spectroscopy was used for monitoring the stability of the complexes (24 h or 72 h) in solution. UV-Vis spectra of **2a** and **2b** were recorded in DMSO and H_2_O. The data showed both complexes were stable in the DMSO solution at the given conditions for 24 h. The obtained scanning kinetics are reported in Appendix A (Appendix A). The electronic spectra in DMSO of the complexes show two intraligand transitions bands (260–276 and 308–346 nm ranges), the absorbance band charge transfer LMCT (379 and 473 nm) and to **2a** d-d transitions in 668 nm. Under the conditions used in the UV-Vis studies, is not observed the d–d transition in the complex **2b**. 

UV-Vis absorption spectra of complexes were monitored and recorded for 72 h in aqueous solutions. The shape of the spectra did not show significant changes with time, as depicted in Appendix A. The decrease in absorbance is observed after two hours and is probably due to precipitation (not visible to the naked eye) under the best conditions. 

#### 3.1.7. Electrochemical Studies

The electrochemical behavior of the compounds was investigated using cyclic voltammetry with ferrocene/ferrocenium couple used as a standard. The results are summarized in Table 4 and voltammograms are presented in Appendix A. 

In the ligands is observed two anodic waves assigned to the irreversible oxidation of the ligand molecules at Epa values of 1.02 and 1.56 V vs. SCE (**1a**), 1.12 and 1.56 V vs. SCE (**1b**). Complexes **2a** and **2b** show quase-reversible redox responses in the positive potential region, attributed to the one-electron transfer [84]. Peak Epa1 is assigned to the oxidation of the central metal of Cu(II) [85]. Peaks Epa2 and Epa3 are observed at nearly the same potential value as the corresponding ligand. Peak Epc1 (catodic processes) is observed at 0.21 V to **2a** and 0,16 V to **2b.** Complex **2a** showed a lower ΔE value than **2b** (240 mV and 356 mV, respectively).

### 3.2. Biological Activities Studies

#### 3.2.1. Determination of Antioxidant Capacity

An imbalance between the production and accumulation of free radicals, such as reactive oxygen species (ROS) and reactive nitrogen species (RNS), causes oxidative stress in cells and tissues. Chain reactions can lead to undesirable side effects in organisms, such as inflammation or cancer [86]. The harmful effects of ROS are balanced by the antioxidant action of antioxidants enzymes and non-enzymatic antioxidants [87].

Antioxidants are substances that can neutralize ROS by donating one of their electrons, protecting cells from damage caused by these species [77]. The antioxidants present two inverse effects on cancer. Some antioxidants positively affect cancer treatment, while others show inducer effects on cancer initiation and progression [88,89]. 

Due to the interesting properties of antioxidants, in this study, we screened the antioxidant activity of free ligands (**1a**, **b**) and copper complexes (**2a**, **b**) using the DPPH and ABTS^+^ assays. The antioxidants assays were carried out using different concentrations (1, 5, 10, 50, 100, and 200 µM) of the compounds test samples. Ascorbic acid, a widely known antioxidant agent, was used as a positive control. The results are presented as % of absorbance control at 517 nm to DPPH (Appendix A) and 730 nm to ABTS^+^ (Appendix A) assays, Table 5.

The 2,2-diphenyl-1-picrylhydrazyl (DPPH^.^) radical assay is commonly used to evaluate the effectiveness of metal complexes containing Schiff bases with antioxidant properties [90,91]. This method was developed in the 1950s [92] and was one of the first assays to be used to assess antioxidant capacity. It is a simple, accurate, and reproducible colorimetric method that is based on the ability of a given antioxidant to reduce the DPPH radical to hydrazine.

From our results, the ligands **1a**, **b** showed an inability to be antioxidants in the DPPH radical capture assay at the test doses (see information support Appendix A).

The DPPH scavenging ability of the test samples of Schiff base metal complexes **2a**, **b** on the basis of percent inhibition is presented in Table 5.

Complex **2a** potentially inhibits the DPPH radical from 5 µM, with values up to 13.9% of inhibition at 50 micromolar and I_max_ of 86.1 ± 1.38%. At the evaluated doses, starting from 100 micromolar, compound **2a** presents a pro-oxidant activity. However, complex **2b**, only at 100 µM showed radical inhibition DPPH of 61.36 ± 4.1%.

IC_50_ values were calculated and compared with the ascorbic acid. IC_50_ is a sufficient concentration to obtain 50% of the maximum scavenging activity. The smaller IC_50_ value corresponds to the greater scavenging activity. Data of the IC_50_ are shown in Table 5. **2a** revealed better antioxidant activity compared with the standard (ascorbic acid) in the DPPH assay (IC_50_ = 6.3 µM to **2a** and 17.0 µM to ascorbic acid). **2b** shows no 50 % inhibition of the radical DPPH, which is taken as evidence of its antioxidant inability in this assay.

In the ABTS^+^ assay, compound **1a** and its complex **2a**, starting from 5 µM, presented activity on scavenging ABTS^+^ radical (Figure 4) with I_max_ values up to 87% at 50 µM. The free ligand **1b** was inactive at all doses, and its complex **2b** inhibited 46% of the ABTS^+^ radical at 200 µM.

The compounds **1a** and **2a** (IC_50_ = 7.5 ± 2.2 to **1a** and 11.8 ± 0.0 to **2a**) demonstrated similar effects on scavenging the ABTS**^+^** radical to ascorbic acid (IC_50_ 10.8 µM), a positive control used in this assay. 

Our studies reveal that compound **2a** presented electron capture activity in the DPPH and ABTS assays, thus showing better antioxidant activity in this study. In the DPPH assay, considering the free ligand **1a** and its **2a** complex, the coordination of the ligand to the metal center is essential for antioxidant activity. In this way, the Cu(II) center, which can be reduced to Cu(I), appears to be involved in the electron transfer mechanism. Ligands **1a** and **1b** have an electron donor unit (–C=N–) system in their structure. Additionally, in structural terms, **1a** presents substituted phenyl and –CH_2_CH_2_– to increase the free radical stabilization and NH group. It has been demonstrated that if the E_HOMO_ is less negative, it indicates that the structure is more unstable and an electron has more affinity to leave the molecule [93]. In our previously discussed theoretical study, in Table 2, **2a** and **1a** have the least negative E_HOMO_. 

One or more of the following mechanisms, HAT, SETPT, and SPLET, are involved in the interaction of free radicals with organic molecules [94]. Among the electronic descriptors calculated and presented in Table 3, IP is the most crucial parameter for the STPT mechanism. In general, lower IP is more favorable to ionization and increases the electron transfer rate between antioxidants and free radicals. Our studies reveal that compound **2a** presented electron capture activity in the DPPH and ABTS assays, thus showing better antioxidant activity in this study. 

#### 3.2.2. Cytotoxicity Tests in V79 Cells Assessment by MTT Assay

The evaluation of cellular cytotoxicity using cell culture assays plays a crucial role as one of the steps in the study of a new drug. To assess whether compounds are cytotoxic in the concentration range that demonstrated activity in scavenging the DPPH and ABTS^+^ radicals, we evaluated the cytotoxicity of compounds against the V79 cell line by MTT assay. This cell line is widely used in toxicity studies because it can keep basal cell functions in normal cell culture conditions [95]. The cytotoxic effect of the ligands **1a**, **b** and complexes **2a**, **b** expressed as cell viability was determined after V79 cells were incubated for 24 h with different concentrations (1, 10, 50, and 100 μM) of the compounds. Figure 5 shows the % viability of V79 fibroblasts after treatment with the compounds. Our results demonstrated that the ligands **1a**, **b** and complexes **2a**, **b** have cytotoxic effects in the V79 cell line in a dose-dependent manner and were quantified in terms of IC_50_ (see Table 6). The lower the IC_50_ value, the higher the cytotoxic effect. Complex **2a** and its free ligand **1a** showed similar IC_50_ values, slightly above the concentrations of these compounds in the antioxidant assays (IC_50_ = 19 ± 1 µM and 23 ± 0 to **2a** and **1a**, respectively). Complex **2b** was the one that presented the highest cytotoxicity, with an IC_50_ = 8 ± 4 µM and its free ligand **1b** was the compound that showed the highest IC_50_ value (69 ± 9 µM).

#### 3.2.3. Study of Cytotoxicity in Human Cell Lines

Considering that the ligands (**1a**, **b**) and their complexes (**2a**, **b**) were cytotoxic against V79 cells at interesting low concentrations, the cytotoxicity evaluation was expanded to human tumor cells. Cytotoxic compounds in tumor cells may play an essential role in cancer treatment studies. Thus, MCF-7 (breast adenocarcinoma), SW620 (colorectal carcinoma), and MRC-5 (normal lung fibroblast) cell lines were exposed to these compounds for 24 h and 72 h.

The cytotoxicity activity was measured by MTT assay and the results are summarized in Appendix A. The free ligands and complexes were cytotoxic on all cell lines in a dose-dependent manner and were quantified in terms of IC_50_. The lower the IC_50_ value, the higher the antiproliferative activity. The half-maximal inhibitory concentration (IC_50_) was calculated to measure the compound’s efficacy. The IC_50_ can be seen in Table 7 and indicates the concentration of the compound needed to inhibit 50% of cell viability. The measurement of the compound selectivity index was found considering the ratio between the cytotoxic parameters found in the MRC-5 cell line (IC_50_ healthy cells) and those observed in tumoral cell lines (IC_50_ MCF-7 or SW620). 

As can be seen in Appendix A, the cytotoxicity of compounds **1a** and **2a** at 24 h and 72 h of treatment starts at 10 µM in all cell lines tested. Compound **1b** showed cytotoxicity starting at 1 µM after 24 h and 72 h of exposition in the SW620 cell line and 10 µM for the MRC-5 and MCF-7 cell lines. Complex **2b** showed very high cytotoxic activity above 30 µM for 24 h and 10 µM for 72 h of the treatment, so it was impossible to measure viability. Therefore, we reduced the test concentrations of this compound, as can be seen in Figure 6. Complexes **2a** and **2b** showed high cytotoxicity at the range 1–100 µM and 1–10 µM, respectively, after 72 h of exposition in the all-cell lines (Appendix A and Figure 6). IC_50_ values to **2a** in the SW620 (IC_50_: 25.5 ± 3.8 μM) and MCF-7 (IC_50_: 30.0 ± 2.9 μM) malignant cell line was higher than in normal cell cells (MRC-5) (IC_50_: 15.4 ± 4.1 μM) after 72 h of exposure (Table 7). Complex **2a** seems not to increase the cytotoxicity of its ligand **1a**, while **2b** has higher cytotoxicity than its ligand **1b**. Table 7 shows that the IC_50_ of **2b** was quite low for MCF-7 (IC_50_: 3.6 ± 0.6 μM), SW620 (IC_50_: 8.3± 1.2 μM), and MRC-5 (IC_50_: 11.0 ± 0.1μM) cell lines for 24 h. The IC_50_ in the SW620 (IC_50_: 4.2 ± 0.3 μM) and MCF-7 (IC_50_: 5.2 ± 0.4 μM) malignant cell lines was similar to MRC-5 cell line (IC_50_: 6.1 ± 0.4 μM) after 72 h of exposure (Table 7). Furthermore, the cytotoxicity of the **2b** against SW620 and MCF-7 cell line at 72 h is close to those found for the reference compound 5-Fluorouracil [(IC_50, 72h_ of 4.2 ± 0.3 μM in SW620 to **2b** and of 4 μM in SW620 to 5-Fluorouracil [96])] and a bit higher than the found for the reference compound cisplatin [(IC_50, 72 h_ of 5.2 ± 0.4 μM in MCF-7 to **2b** and of 12.7 ± 0.1 μM in MCF-7 to cisplatin [97])], respectively. 

When comparing the IC_50_ value of the compounds between the normal and malignant cell lines, it is noted that the compounds do not show selectivity for tumor cells (MCF-7 and SW620) in this assay. 

#### 3.2.4. In Vitro Colony Survival Assay

Clonogenic assay was used to confirm the inhibition activity towards human cell lines MCF-7, SW620, and MRC-5 of the ligands (**1a**, **b**) and Cu(II) complexes (**2a**, **b**). The clonogenic assay is based on the ability of a single cell to grow in a colony, being more reliable in evaluating the selectivity of each compound in cell proliferation. Inhibition of cell proliferation can occur by inducing death (apoptosis, necrosis, autophagy), by mitotic catastrophe, or even by senescence [98]. 

MRC-5, SW620, and MCF-7 cells were treated with different compound concentrations for 72 h and then cultured in the medium for 8, 10, and 14 days of culture, respectively, to allow colony growth. The results showed a concentration-dependent reduction in the number of colony-forming cells by the complex (Figure 7 and Appendix A). In this assay, the results indicate that all compounds, except **1a**, show selectivity for a tumor cell line in one of the concentrations. Compound **2a** inhibited the proliferation of tumor cell lines more than the MRC-5 normal cell line at 20 µM, while its ligand was not selective. Compound **2b** showed a more significant inhibitory effect on the MCF-7 cells than in MRC-5 cells, while its ligand showed a more potent inhibitory effect on the SW620 cells than on MRC-5 cells. Statistical data also indicate that all compounds significantly inhibited cell proliferation at the highest treatment concentrations tested, except for compound **1b** in the MRC-5 cells, where the inhibition was not statistically significant (Appendix A).

#### 3.2.5. Genotoxicity Effects in Human Cell Lines

Considering the cytotoxic effects observed we investigated if the **1a**, **1b**, **2a**, and **2b** compounds were able to induce DNA damage (i.e., single (SSBs) and double-strand breaks (DSBs), alkali-labile sites, DNA adducts, excision repair sites, and cross-links). As can be seen in Table 8, the **1a** compound increases damage index (DI) at concentrations of 10–100 µM in MRC-5 and MCF-7 cell lines, as well as 50 µM and 100 µM in the SW620 cell line. The compounds **2a** and **1b** increase DI at 50 µM and 100 µM in MRC-5 and MCF-7 cell lines, as well as ≥10 µM in the SW620 cell line. The **2b** compound increased DI in low concentrations, at concentrations ≥4 µM in MCF-7 and SW620 cell lines. In fact, DI for **2b** compound at 10 µM concentration was about three times higher in MCF-7 cells than in MRC-5 cells. Further studies with **1a**, **b** and **2a**, **b** compounds should be performed to provide a better understanding of the mechanisms underlying the cytotoxic and genotoxic effects.

#### 3.2.6. Molecular Docking 

All compounds were able to induce DNA damage. Cu^2+^ is a redox-active metal ion and can cause oxidative DNA damage and strand breaks. Furthermore, Cu^2+^ ions are capable of binding coordinatively with nuclear proteins and DNA causing site-specific damage [99]. This redox state of copper ion coordinates with the phosphate groups and nucleobases. Cu(II) complexes can additionally bind to DNA via non-covalent interactions that are governed by the coordination geometry and the nature of the ligand chelated in the metal center. If these ligands carry additional groups for hydrogen bonding, electrostatic, hydrophobic, or π–π stacking interactions, they can be used to enhance the binding affinity or to confer selectivity and target recognition of specific DNA sequences, conformations or higher-order structures [100].

Docking calculations of complexes **2a** and **2b** were performed to analyze if these compounds are DNA-targeting agents. Results from docking simulations show that the compounds have similar affinity values for DNA, of 6.9 and 7.4 kcal mol^−1^ for **2a** and **2b**, respectively. Both complexes bind to the minor groove of DNA (Figure 8). Complex **2a** showed hydrophobic and polar interactions of the charged amine moiety with DNA, but complex **2b** showed more favorable π-interactions of the aromatic amine moiety of the complex with residues guanine and cytosine in DNA. This fact illustrates the importance of the conjugated, fused ring of **2b** as compared to **2a**. Furthermore, complex **2b** shows an alternative mode of binding with the same energy (7.4 kcal mol^−1^), where the compound is bonded to the major groove of DNA (Appendix A), with the rings parallel to the double helix. This orientation favors several π-interactions of the complex with the DNA. The presence of an extended aromatic ring in **2b** favors the interaction of this complex with DNA and a presence of a hydrogen donor, charged amine in **2a** favors its interaction with DNA.

## 4. Conclusions

In this study, two new copper complexes chelated by an amine moiety N-donor, Schiff base chromophore, and phenoxide fragments have been developed. Single crystal X-ray studies for **2a** show a mononuclear and dinuclear Cu(II) complex with the Cu-Cl bond distance indicating weak interaction between the chloride and the copper center. In both cases, the copper centers are chelated by one K^3^-O,N,N-phenoxy-imine-amine ligand. The copper center is tetracoordinated with a distorted quadratic geometry in the mononuclear species. DFT calculations showed that the energy difference (ΔE) between the dinuclear and mononuclear complexes is only −7.42 kcal^.^mol^−1^. Furthermore, DFT calculations established the optimized structure of ligands and their corresponding Cu(II) complexes. The findings revealed a good correlation between experimental and theoretical data. The ESI-HRMS results indicated that the monomeric nature of **2a** and **2b** species is maintained in solution. The electrochemical behavior of ligands and their Cu(II) complexes were studied.

Our study demonstrated that complex **2a** has shown better antioxidant activity. All compounds had cytotoxicity and genotoxic effects under tissue culture conditions, and complex **2b** induced about three times higher genotoxicity in MCF-7 cells than in MRC-5 cells at a concentration of 10 µM. Thus, low **2b** doses may be useful in the development of adjuvant therapies or rational combinations that may be predicted to have synergistic or additive effects in combination with currently used chemotherapeutics. Our in silico analysis sheds light at the molecular level on potential **2a**, **b**-modulated interactions with DNA. These compounds exhibit a high degree of interaction with DNA due to the presence of an extended aromatic ring in **2b**. In addition, they also have an amine charged with a hydrogen donor in **2a**, demonstrating that these compounds have great potential for biomedical applications.

## Data Availability

The data presented in this study are available in this article. Crystallographic data for the structure in this work were deposited in the Cambridge Crystallographic Data Centre as supplementary publication number CCDC 2203554. Copies of the data can be obtained free of charge via www.ccdc.cam.ac.uk/data_request/cif (accessed on 21 December 2022).

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
