# Peer review of "Antioxidant and Anticancer Potential of the New Cu(II) Complexes Bearing Imine-Phenolate Ligands with Pendant Amine N-Donor Groups"

_pharmaceutics, 2023, doi:10.3390/pharmaceutics15020376_

Round 1

Reviewer 1 Report

Dear Authors,

please follow the comments and suggestions and correct the article.

Author Response

We would like to thank the suggestions and comments, which certainly improved our manuscript. The reviewers´ comments encouraged us to clarify all the raised concerns. Hence, the text and figures were changed in response to the questions. In this sense, we have re-written parts of the manuscript to address all comments, highlighting the corresponding changes in yellow in the manuscript body.

Minor review:

After carefully considering your comments, we have modified our text according to their suggestions. Each of your specific comments has been marked up using the “Track Changes” function, and all changes made to the manuscript were highlighted.

Major review:

  1. [M-Cl]+ seems to be the correct description. But the obtained value of 415.1337 quite strongly differs from the calculated one (415.1811). 

High‐resolution mass spectra (HRMS) of 2a was obtained (Please see page 3), “ESI-HRMS (MeOH, m/z) = 414.1711 [M – Cl]+ (calcd for C23H31CuN2O: 414.1732)”.

  1. Ligand 1a was prepared as described in [50] according to the procedure described in one of…

[6-(Ph(NH)-C2H4-(N=CH)-2,4-di-tert-butyl-(OHC6H2)] (1a), and [6-(C9H6N-8-(N=CH))-2,4-di-tert-butyl-(OHC6H2)] (1b) ligands were prepared according to the procedure described in [49, 50] (Pease see page 3).

  1. DFT calculations showed that the energy difference (ΔE) between the dinuclear (2a’) and mononuclear (2a) complexes is only -7.41 kcal×mol-1 [2a’: E = -3990.259 kcal×mol-1 1; 2a: E = -3999.251 kcal×mol-1]. How the energy difference was calculated?

“DFT calculations showed that the energy difference (ΔE) between the dinuclear (2a’) and mononuclear (2a) complexes is only -7.42 kcal×mol-1 [2a’: E = -3990259.34 kcal×mol-1; 2a: E = -3990251.92 kcal×mol-1].”(Please see page 10). (2a’ – 2a)

  1. DI for MCF-7 is 252, and for MRC-5 reaches 75. Is this value 5-fold lower than for MCF-7?

“DI for 2b compound at 10 µM concentration was about three times higher in MCF-7 cells than in MRC-5 cells.” (Please see page 20)

Reviewer 2 Report

1. In the bioactivity assay, 2b showed more significant inhibitory effect on the MCF-7 cell lines than on MRC-5. Why chose MRC-5 as positive control?

2. In vitro data are usually inconsistent with in vivo data. Authors should analyze animal study to evaluate the real in vivo anticancer effects of 2b.

3. Comparison of the antitumor activity of clinical anticancer drug and 1a to 2b should be provided.

4. Page 17, the line spacing is different than former part.

IC50 values to 2a in the SW620 (IC50: 25.5 ± 3.8 μM) and MCF-7 (IC50: 30.0 ± 2.9 μM) cancer cell line was higher than in human fibroblast cells (MRC-5) (IC50: 15.4 ± 4.1 μM) after 72 h of exposure (Table 7). Complex 2a seems not to increase the cytotoxicity of its ligand 1a, while 2b has higher cytotoxicity than its ligand 1b. Table 7 shows that the IC50 of 2b was quite low for MCF-7 (IC50: 3.6 ± 0.6 μM), SW620 (IC50: 8.3± 1.2 μM), and MRC-5 (IC50: 11.0 ± 0.1μM) cell lines for 24 h. The IC50 in the SW620 (IC50: 4.2 ± 0.3 μM) and MCF-7 (IC50: 5.2 ± 0.4 μM) cancer cell lines was similar to human fibroblast cells (MRC-5) (IC50: 6.1 ± 0.4 μM) after 72 h of exposure (Table 7). When comparing the IC50 value of the compounds between the cell lines, it is noted that the compounds do not show selectivity for tumor cells (MCF-7 and SW620) in this assay.

Author Response

We would like to thank the suggestions and comments, which certainly improved our manuscript. The reviewers´ comments encouraged us to clarify all the raised concerns. Hence, the text and figures were changed in response to the questions. In this sense, we have re-written parts of the manuscript to address all comments, highlighting the corresponding changes in yellow in the manuscript body.

  1. In the bioactivity assay, 2b showed more significant inhibitory effect on the MCF-7 cell lines than on MRC-5. Why chose MRC-5 as positive control?

The MRC-5 cell line was not used as a positive control but as a non-malignant cell line to compare the effects of the ligands and complexes with the different malignant cell lines used in the study, with different response profiles, and thus to see if there was any selectivity or resistance for future studies. To reinforce and clarify the use of the MRC-5 cell line, we have added sentences in the section you commented on and throughout the paper (All changes made to the manuscript were highlighted). On section 3.2.3. Study of Cytotoxicity in Human Cell Lines you can find the answer.

“” The measurement of the compound selectivity index was found considering the ratio between the cytotoxic parameters found in the MRC-5 cell line (IC50 healthy cells) and those observed in tumoral cell lines (IC50 MCF-7 or SW620).’’ (Please see page 17)

‘’Indeed, when comparing the IC50 value of the compounds between the normal and malignant cell lines, it is noted that the compounds do not show selectivity for tumor cells line (MCF-7 and SW620) in this assay.’’ (Please see page 17)

  1. In vitro data are usually inconsistent with in vivo data. Authors should analyze animal study to evaluate the real in vivo anticancer effects of 2b.

Thank you for your suggestion. We agree that the results of the in vivo model can provide relevant data to our manuscript, and this certainly will be done in the next steps.

As the reviewer also knows is that the experiments with cell lines are also very necessary in pre-clinical studies as a first screening of compounds and therefore to reduce the number of animals in the experiments.  

. In addition, it is not feasible to conduct an in vivo study in a timely manner, considering the time frame for the response (10 days).

  1. Comparison of the antitumor activity of clinical anticancer drugs and 1a to 2b should be provided.

Thank you for your suggestion. On section 3.2.3. Study of Cytotoxicity in Human Cell Lines you can find the answer.

“The IC50 in the SW620 (IC50: 4.2 ± 0.3 μM) and MCF-7 (IC50: 5.2 ± 0.4 μM) malignant cell lines was similar to MRC-5 cell line (IC50: 6.1 ± 0.4 μM) after 72 h of exposure (Table 7). Furthermore, the cytotoxicity of the 2b against SW620 and MCF-7 cell line at 72h is close those found for the reference compound 5-Fluorouracil [(IC50, 72h of 4.2 ± 0.3 μM on SW620 to 2b and of 4 μM on SW620 to 5-Fluorouracil [96])] and a bit higher than the found for the reference compound cisplatin [(IC50, 72 h of 5.2 ± 0.4 μM on MCF-7 to 2b and of 12.7 ± 0.1μM in MCF-7 to cisplatin [97])], respectively.” (Please see page 17)

  1. Page 17, the line spacing is different than former part.

IC50 values to 2a in the SW620 (IC50: 25.5 ± 3.8 μM) and MCF-7 (IC50: 30.0 ± 2.9 μM) cancer cell line was higher than in human fibroblast cells (MRC-5) (IC50: 15.4 ± 4.1 μM) after 72 h of exposure (Table 7). Complex 2a seems not to increase the cytotoxicity of its ligand 1a, while 2b has higher cytotoxicity than its ligand 1b. Table 7 shows that the IC50 of 2b was quite low for MCF-7 (IC50: 3.6 ± 0.6 μM), SW620 (IC50: 8.3± 1.2 μM), and MRC-5 (IC50: 11.0 ± 0.1μM) cell lines for 24 h. The IC50 in the SW620 (IC50: 4.2 ± 0.3 μM) and MCF-7 (IC50: 5.2 ± 0.4 μM) cancer cell lines was similar to human fibroblast cells (MRC-5) (IC50: 6.1 ± 0.4 μM) after 72 h of exposure (Table 7). When comparing the IC50 value of the compounds between the cell lines, it is noted that the compounds do not show selectivity for tumor cells (MCF-7 and SW620) in this assay.

Thank you very much for your comments, it has been corrected in the text.

Reviewer 3 Report

The article sent to me for review is exceptionally good, well prepared and concerning an important and interesting therapeutic problem.

As can be seen from the recently published numerous research papers, the observation of the correlation of antioxidant and anticancer properties is an extremely promising direction in the search for new therapeutic methods and new pharmaceutics. The use of copper (II) ions is an equally accurate idea, mainly due to their relatively friendly nature towards the human body. So the concept of work is very good and well justified. The scope of the experimental research carried out is also highly appreciated. Their method of carrying out and the way of presenting the results remain without any major reservations. And although the article concerns practically only two complex compounds and two of their precursors, it is prepared for them in an almost exhaustive way. With the presented scope of all research, the description of the docking of molecules to the receptor and the related calculations is somewhat modest. At this point, I feel a certain deficiency. But it does not need to be supplemented, because the presented results in combination with all other data are completely sufficient to conduct a proper discussion and draw appropriate conclusions. The authors coped very well with the division of the presented material into the basic part of the work and separate supplementary materials. They are equally well prepared and well documented the overall work.

Among others, the title of the Table 5 is given in the authors' native language. References are edited in a way that deviates from the convention adopted in the MDPI Publishing House. In many places, the work requires font correction, spaces, justification and similar corrections. The latter can be introduced at the stage of editorial correction, when the text will be preparing for edition. 

Author Response

We would like to thank the suggestions and comments, which certainly improved our manuscript. The reviewers´ comments encouraged us to clarify all the raised concerns. Hence, the text and figures were changed in response to the questions. In this sense, we have re-written parts of the manuscript to address all comments, highlighting the corresponding changes in yellow in the manuscript body.

Among others, the title of the Table 5 is given in the authors' native language. References are edited in a way that deviates from the convention adopted in the MDPI Publishing House. In many places, the work requires font correction, spaces, justification and similar corrections. The latter can be introduced at the stage of editorial correction, when the text will be preparing for edition. 

Thank you very much for your comments, it has been corrected in the text.

Round 2

Reviewer 1 Report

Dear Authors,

you manuscript needs a minor revision. 

Manuscript comments:

P2: … antimicrobial activity [43, 44]. …, because [43] and [44] reports antimicrobial activity.

P6: section 2.7.5., L4: ...described by Singh et al. [73] Following... please correct it.

P21: Table 8 needs to be reuploaded.

P24: The Reference List should be carefully checked and submitted in
accordance with the journal's guidelines for citing sources (
Journal Articles:
1. Author 1, A.B.; Author 2, C.D. Title of the article. Abbreviated Journal
Name
 YearVolume, page range). Please note the providing of journal title
abbreviation
and that the volume of the journal must be given.
If you provide both the volume and the issue, please check and correct
it in the list, because, some issues are missing in the list.

Author Response

We would like to thank the comments.

Point-by-point response to reviewers 

Minor review:

  1. P2: … antimicrobial activity [43, 44]. …, because [43] and [44] reports antimicrobial activity.

We have modified our text according to suggestions (please see page 2).

  1. P6: section 2.7.5., L4: ...described by Singh et al. [73] Following... please correct it.

Done (please see page 6). 

  1. P21: Table 8 needs to be reuploaded. 

Done (please see page 20).

  1. P24: The Reference List should be carefully checked and submitted in accordance with the journal's guidelines for citing sources (Journal Articles: 1. Author 1, A.B.; Author 2, C.D. Title of the article. Abbreviated JournalName Year, Volume, page range). Please note the providing of journal title abbreviation and that the volume of the journal must be given. If you provide both the volume and the issue, please check and correct it in the list, because, some issues are missing in the list.

 Done

Reviewer 2 Report

Accept your revised.

Author Response

We would like to thank the suggestions and comments, which certainly improved our manuscript. 
